# Mouse microglia express unique miRNA-mRNA networks to facilitate age-specific functions in the developing central nervous system

Alexander D. Walsh[1,6], Sarrabeth Stone[1], Saskia Freytag[2,3], Andrea Aprico[1], Trevor J. Kilpatrick[1], Brendan R. E. Ansell [3,4,7] & Michele D. Binder [4,5,7 ✉]

Microglia regulate multiple processes in the central nervous system, exhibiting a considerable level of cellular plasticity which is facilitated by an equally dynamic transcriptional environment. While many gene networks that regulate microglial functions have been characterised, the influence of epigenetic regulators such as small non-coding microRNAs (miRNAs) is less well defined. We have sequenced the miRNAome and mRNAome of mouse microglia during brain development and adult homeostasis, identifying unique profiles of known and novel miRNAs. Microglia express both a consistently enriched miRNA signature as well as temporally distinctive subsets of miRNAs. We generated robust miRNA-mRNA networks related to fundamental developmental processes, in addition to networks associated with immune function and dysregulated disease states. There was no apparent influence of sex on miRNA expression. This study reveals a unique developmental trajectory of miRNA expression in microglia during critical stages of CNS development, establishing miRNAs as important modulators of microglial phenotype.

[1] The Florey Institute of Neuroscience and Mental Health, Parkville, Melbourne, VIC 3052, Australia. [2] Personalised Oncology Division, The Walter and Eliza Hall Institute of Medical Research, Parkville, VIC 3052, Australia. [3] Department of Medical Biology, University of Melbourne, Parkville, VIC 3052, Australia. [4] Population Health and Immunity Division, The Walter and Eliza Hall Institute of Medical Research, Parkville, VIC 3052, Australia. [5] Department of Anatomy and Physiology, University of Melbourne, Parkville, Melbourne, VIC 3052, Australia. [6] Present address: Cognitive Neuroepigenetics Laboratory, Queensland Brain Institute, The University of Queensland, Brisbane, QLD, Australia. [7] These authors jointly supervised this work: Brendan R. E. Ansell, Michele D. Binder. ✉email: mbinder@florey.edu.au

Microglia are dynamic regulators of the central nervous system (CNS) where they support early development, adult homoeostasis and immune function. Following embryonic colonisation of the brain, microglia regulate neurogenesis, synaptogenesis and myelination via both the clearance (efferocytosis) of immature cells and synapses, and secretion of trophic factors[1–4]. In the adult brain, microglia continue to regulate existing neural networks, as well as supporting neurogenic niches and oligodendrocyte precursor cell pools[5,6]. Adult microglia adopt a ramified phenotype and extend processes to survey the local environment and maintain tissue homoeostasis[7,8]. Upon interaction with pathological stimuli, microglia can rapidly shift their phenotype to initiate apoptotic clearance and cytokine signalling to suppress inflammation and promote a neuroprotective environment[9].

Given the broad spectrum of phenotypes and activities of microglia in the healthy CNS, appropriate microglial function is heavily dependent upon precise regulation of gene expression. Consequently, genetic dysregulation of microglial biology is implicated in numerous neurological disorders. Chronically activated 'neurotoxic' microglia have been identified as a hallmark of neurodegeneration and autoimmunity, with key genes and signalling pathways explicitly linked to the pathology of multiple sclerosis (MS), amyotrophic lateral sclerosis (ALS), Alzheimer's disease (AD) and Huntington's disease (HD)[10]. Perturbed microglial activity is also implicated in neurodevelopmental disorders including autism spectrum disorder (ASD), schizophrenia and epilepsy[11–14].

Multiple sequencing studies of microglial populations have captured genetic and epigenetic networks that regulate cell identity and define microglial phenotypes in the healthy and diseased brain[15–19]. In addition, comprehensive profiling has identified distinct programmes of gene expression that tightly regulate developmental phenotypes and age-specific microglial functions[20,21]. Pre-natal and early postnatal microglia are highly reactive and adopt an amoeboid morphology similar to that of adult-activated microglia during inflammation and disease. However, there is no strong gene expression overlap between these cell subsets, indicating that developmental microglia are a distinct cell population that uniquely contribute to CNS development. These studies highlight the multiplicity of transcriptional programmes which can be activated by microglia in response to normal development or pathological challenge. An important question then arises as to how these transcriptional programmes are controlled? One strong potential candidate is the class of small RNAs known as microRNAs (miRNAs).

miRNAs are small (18–22 bp) non-coding RNAs that act as negative post-transcriptional regulators of gene expression[22]. Mature miRNAs are integrated into a catalytic complex that binds to target mRNAs and triggers degradation or stalled translation of the target mRNA transcript[23]. Ubiquitous in the mammalian genome, miRNAs regulate fundamental processes including cell differentiation, proliferation and immune homeostasis[24,25]. Tissue-specific sequencing of miRNAs has revealed that ubiquitously expressed miRNAs are less common than previously predicted and emphasise the importance of cell-specific studies to capture accurate miRNA profiles[26]. Normal specification of microglia is known to be reliant upon miRNAs. Conditional knockout of Dicer in microglia ablates the miRNA biogenesis pathway, which results in perturbation of microglia and induces hyper-responsiveness to inflammatory stimuli[27]. In addition, specific miRNAs have a profound influence on microglial biology. For example, miR-155 and miR-124 have been identified as 'master regulators' of activated and quiescent microglia, respectively[28–30]. However, much of the evidence for miRNA-mediated regulation of microglia stems from studies of adult

mice, and thus the role of miRNAs in regulating developmental microglial functions are not as well understood. Further, studies of the miRNAome are often limited in their ability to predict transcriptional effects, as they rely on in silico predictions of miRNA–mRNA interactions rather than integrated network analyses.

In this study, we have characterised the miRNAome of microglia in male and female mice at three developmental timepoints: postnatal days 6, 15 and 8 weeks. These timepoints represent critical stages in CNS development in the mouse. We identified a unique microglial miRNA profile compared with other CNS cell types. Differential expression analysis identified age-specific subsets of miRNA expression, suggesting a gene regulatory programme that influences microglial biology during development. Unexpectedly, we did not observe any sex-specific differential miRNA expression, and found only a modest difference in mRNA expression, indicating that microglia are not necessarily innately different between sexes during development. Integrated miRNA–mRNA expression analyses identified over 200 mRNAs that were negatively correlated with enriched microglial miRNAs, and that were predicted to target key neurodevelopmental processes, including neurogenesis, axonal guidance and myelination. These data provide a unique profile of the microglial transcriptome and its epigenetic regulation, providing insight into the role of microglial miRNAs in regulating microglial developmental programmes.

## Results

**Isolation of highly pure microglial populations via CD45$^{+ve}$ immunopanning.** To explore the role of microglial enriched miRNAs and their influence on gene regulatory networks during CNS development, we sequenced and analysed the miRNA and mRNA transcriptomes of CD45$^{+ve}$ microglia isolated from three age groups of C57Bl/6 mice (P6, P15, 8 weeks) as well as the miRNA expression of the CD45$^{-ve}$ bulk CNS cells (primarily comprised of neurons, oligodendrocytes and astrocytes) (Fig. 1). qPCR expression analyses demonstrated that microglial-specific markers such as Tmem119 and Cx3cr1 were highly expressed by purified CD45$^{+ve}$ cells (Supplementary Fig. 1a). In addition, because CD45 is expressed by non-microglial cells, including monocytes and macrophages, we further assessed the purity of immunopanned samples with CIBERSORTx. RNA-seq gene expression was compared to single-cell microglial datasets generated across the mouse lifespan[20]. CIBERSORTx analysis indicated an average of 2.8% non-microglial (monocyte/macrophage) contamination per sample (Supplementary Fig. 1b).

Multidimensional scaling (MDS) analysis of miRNA sequencing libraries identified strong separation of sample populations by both cell type (microglia vs CD45$^{-ve}$ bulk) as well as age (Fig. 2a). Overall, 1073 unique miRNAs were expressed across all mice, with a minimum count per million (CPM) of 1 in at least 5 samples. Sequencing of microglial mRNA transcriptomes identified 15,660 robustly expressed transcripts (minimum CPM of 0.5 in at least five samples), and MDS analysis revealed a similarly strong separation of samples by age (Fig. 2b). Amongst the most highly expressed microglial miRNAs and mRNAs were genes previously associated with microglial identity and homeostasis, including Cx3cr1, Csf1r, Hexb, Sparc, miR-124, let-7 and miR-29b (Supplementary Table 1).

**Developmental microglia express a common miRNA signature and subsets of temporally dynamic miRNAs.** A primary aim of this study was to characterise miRNAs enriched in microglia relative to other CNS cell types. To do this, we first undertook a general miRNA enrichment analysis (pooling and comparing all

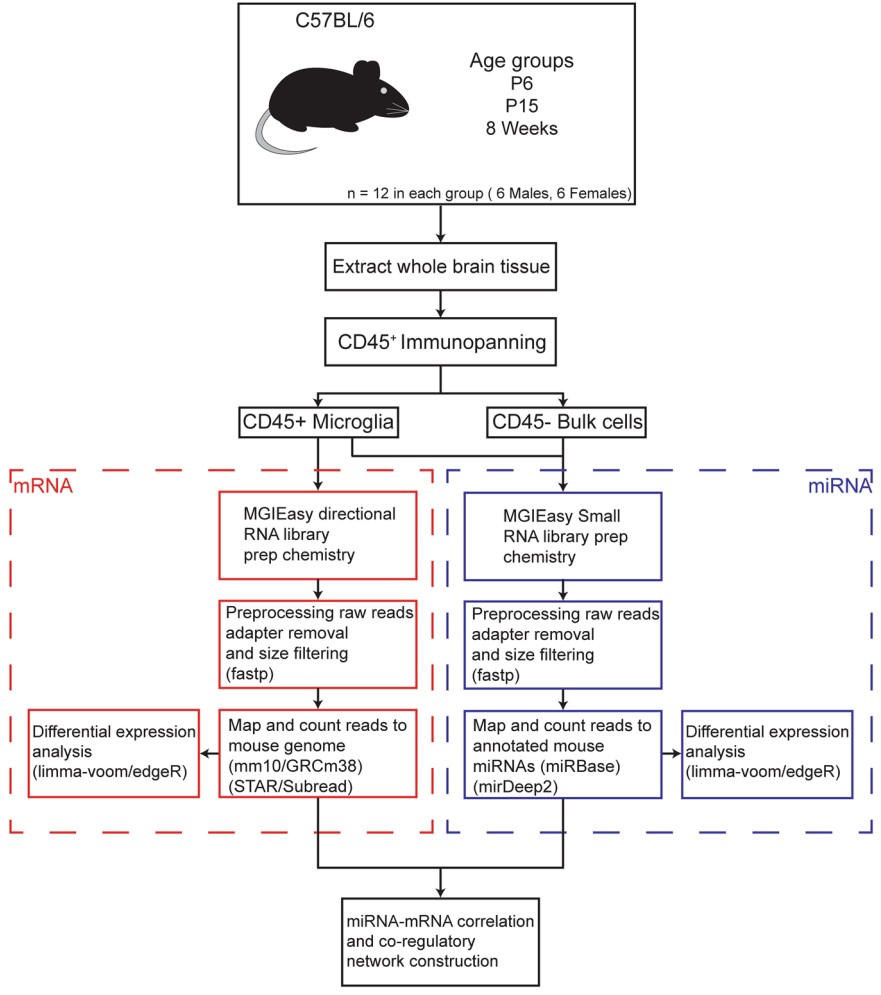

**Fig. 1 miRNA/mRNA sequencing and analysis workflow.** Whole brains of three groups of C57Bl/6 mice ($n = 12$) were processed and CD45$^{+ve}$ microglia were isolated via immunopanning. Microglia and CD45$^{-ve}$ bulk cell populations were sequenced for miRNA expression and analysed for differential expression using the edgeR/limma-voom pipeline. In addition, mRNA expression was sequenced in microglial populations and normalised read counts were correlated with miRNA data to construct miRNA–mRNA coregulatory networks.

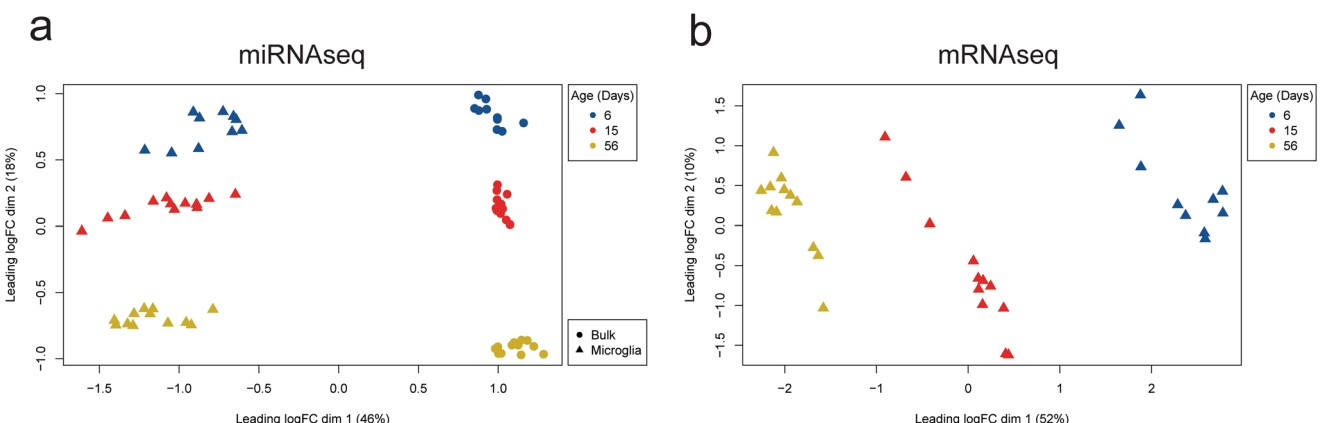

**Fig. 2 MDS analysis of isolated samples identifies strong clustering by sample age and cell type status.** MDS clustering of **a** miRNA sequencing data (CD45$^{+ve}$ microglia and CD45$^{-ve}$ bulk samples) and **b** mRNA sequencing data (microglia only) were performed for 34 mice. miRNA samples clustered by both age (in days) and cell type. Similarly, mRNA sequenced microglia mRNA samples were strongly separated by developmental age.

CD45$^{+ve}$ microglial samples to all CD45$^{-ve}$ bulk samples, controlling for age, sex and mouse ID) which identified 213 miRNAs as upregulated in microglia (Fig. 3a). The most highly enriched miRNAs included many that were previously defined as

important in regulating microglial identity, including miR-142a-5p, miR-223-3p and miR-146a-3p. Also identified as highly enriched were miR-1895, miR-511-3p and miR-6983-5p, which have less well-defined functions in microglia (Fig. 3a). Next, we

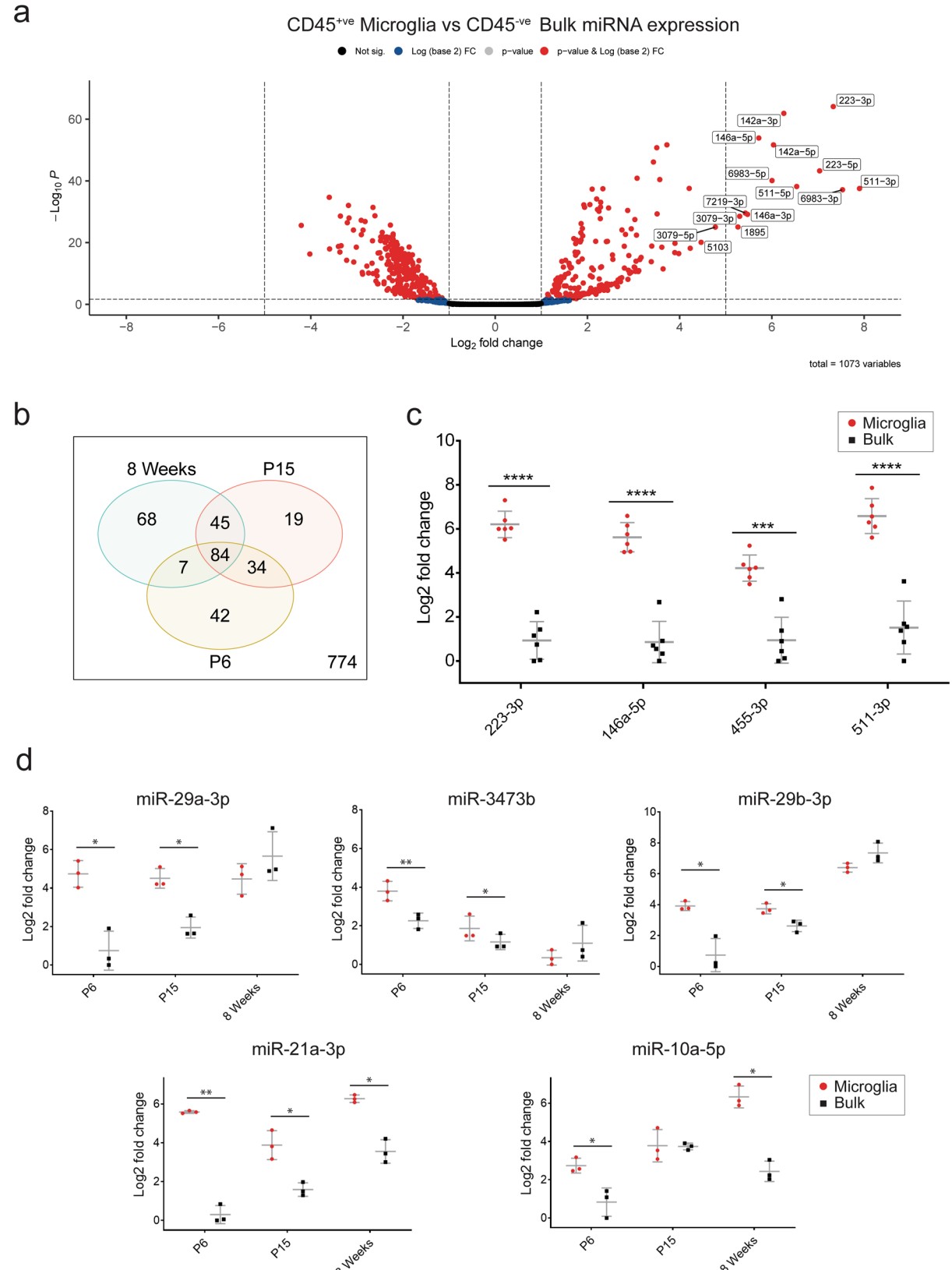

performed the enrichment analysis within each age group to generate age-specific miRNAomes (Supplementary Data 1). Two hundred and ninety-nine miRNAs were defined as enriched in at least one age group, with 84 enriched at all ages, indicative of a common signature of microglial miRNA expression (Fig. 3b). The top ten enriched miRNAs in each age group are presented in Supplementary Table 2. Conversely, 129 miRNAs were enriched in a single age group, with 68 of these specific to adult cells (Fig. 3b). To validate expression of top enriched candidates, we performed qPCR of selected miRNAs, confirming strong upregulation of four miRNAs in 8-week-old mice; miR-223, miR-146a-5p, miR-455-3p, miR-511-3p (Fig. 3c). We extended this

**Fig. 3 Microglia express unique developmental profiles of miRNA expression compared to surrounding CD45$^{-ve}$ bulk cells. a** Volcano plot of differential miRNA expression between pooled microglia samples and CD45$^{-ve}$ bulk samples correcting for age and sex. 213 of 1073 miRNAs were observed to be upregulated in microglia (absolute log$_2$FC > 1, FDR < 0.05). Top 15 upregulated miRNAs (by log2 fold change) in microglia are labelled. **b** Venn diagram comparing subsets of enriched miRNAs (microglia vs CD45$^{-ve}$ bulk) in each age group (absolute log$_2$FC > 1, $P$ < 0.05). Two hundred and ninety-nine miRNAs were observed as upregulated in at least one age group, with 84 consistently upregulated at every age. **c** qPCR validation of selected miRNA candidates in 8-week-old mice ($n$ = 6 in each group). Data were analysed by paired $t$ tests comparing microglia and CD45$^{-ve}$ bulk group expression for each tested miRNA. **d** qPCR validation of selected miRNA candidates across three age groups, P6, P15 and 8 weeks ($n$ = 3 in each group). Data were analysed by paired $t$ test comparing microglia and CD45$^{-ve}$ bulk expression within each age group. *$P$ < 0.05, **$P$ < 0.01, ***$P$ < 0.001, ****$P$ < 0.0001. qPCR data are presented as mean ± SD.

analysis via qPCR at P6, P15 and 12 weeks, observing consistent expression of these miRNAs across microglial development and into adulthood (Supplementary Fig. 2). Extended qPCR analysis also validated specific patterns of observed temporal miRNA expression, including examples of candidates enriched across all ages (miR-21a-3p), enriched specifically at P6 and P15 (miR-29a-3p, miR-29b-3p, miR-3473b) and exhibiting increased expression (and enrichment) throughout development (miR-10a-5p) (Fig. 3d). Overall, this enrichment analysis identified commonly enriched as well as age-specific miRNAs that contribute to the developmental miRNAome of microglia.

**mRNA–miRNA regulatory network analysis identifies age-specific microglial biology that contributes to CNS development.** To characterise the gene regulatory networks potentially regulated by age-specific miRNAomes in microglia, we correlated the expression of microglial-enriched miRNAs and all expressed mRNA transcripts from isolated microglia populations (Figs. 4–6). Given that miRNAs are canonically negatively correlated with their mRNA targets, we selected putative miRNA–mRNA pairs that were strongly negatively correlated ($R^2$ > 0.8, FDR < 0.05). miRNA–mRNA pairs were further filtered for experimental evidence of interaction as reported in the existing literature and relevant databases (collated by multiMiR) and constructed into 'high confidence' miRNA–mRNA regulatory networks for each age group (Figs. 4a, 5a and 6a). We measured the expression of several mRNA targets represented in these networks via qPCR, validating their significant negative correlations (Supplementary Fig. 3). Overall, 24 miRNAs and 234 target mRNA genes were represented in the constructed networks.

A consistent observation for networks was a high degree of interconnectedness, with many genes being targeted by more than one miRNA (Figs. 4a, 5a and 6a and Supplementary Data 2). Ten miRNAs were present in all age group networks, creating a large overlap in represented pathways. Conversely, 6 miRNAs were present in a single network; miR-24-3p, miR-152-3p, miR-27a-3p, miR-92a-3p and miR-10a-5p at 8 weeks and miR-210-3p at P6 (Figs. 4a and 6a). KEGG pathway enrichment analysis of target mRNAs in each network identified strong representation of biological processes related to cell cycle, neurogenesis, axon guidance and pathways including p53, Hippo, PI3K-Akt and JAK-STAT signalling (Figs. 4b, 5b and 6b and Supplementary Data 3). Gene ontology (GO) analysis of target genes in these networks identified enrichment of genes related to 'myelin sheath' across the three networks (Supplementary Data 4). Specific miRNA targets of myelination genes included *Padi2* (regulated by miR-3473b) and *Hmgcr* (regulated miR-29a-3p). In addition, pathways related to neurodegenerative disease were overrepresented in all networks. These age-specific regulatory networks were primarily composed of miRNAs that were enriched in all age groups, and therefore define robust regulation of conserved microglial transcriptional networks during CNS development and adult homoeostasis.

**Subsets of temporally dynamic miRNAs regulate specific aspects of microglial development.** Microglia undergo major phenotypic and transcriptional transitions from early postnatal development through to adulthood[21]. In order to identify the miR-NAs that are most likely to regulate developmental transitions in microglia, we next interrogated changes in miRNA expression between age groups. We performed direct pairwise comparisons of miRNA expression profiles between different development ages to define subsets of differentially expressed miRNAs (Fig. 7a and Supplementary Data 5). One hundred and eighty-five differentially expressed miRNAs were observed across all comparisons, with the largest difference observed between P6 and 8-week microglia, reflecting the maturation of developmental microglia to their adult homeostatic counterparts (Fig. 7b). To further characterise the individual developmental trajectories of differentially expressed miRNAs, we plotted fold changes in expression at P15 and 8 weeks, relative to P6 and categorised significantly altered miRNAs into groups based on their changing patterns of expression (Fig. 7c). This analysis identified 14 miRNAs that were sequentially upregulated, and 37 miRNAs that were sequentially downregulated between P6 and adult microglia (Fig. 7c, d). We constructed miRNA–mRNA networks for these subsets as described above. Gene ontology (GO) analysis of the predicted target genes for the upregulated miRNA subset were strongly related to cyclin activity and cell cycle transition consistent with the proliferative nature of early postnatal microglia[31] (Fig. 7e). In contrast, predicted gene targets of the downregulated miRNA subset were overrepresented in biological processes related to wound healing, endocytosis, migration and immune function (Fig. 7e).

**Analysis of novel miRNAs enriched in microglia identifies chr3_29427 as a putative novel regulator of microglial biology.** Of the 1073 miRNAs detected across all samples, 54 were identified as previously unannotated novel miRNA sequences by miRDeep2[32]. Twenty-four of these novel sequences were specifically enriched in microglia relative to CD45$^{-ve}$ CNS cells (Fig. 8a and Supplementary Table 3). The putative precursor miRNA structures for these sequences are reported in Supplementary Table 4). The most strongly expressed novel miRNA candidate in this regard was chr3_29427, which mapped to an intergenic region on chromosome 3 (chr3: 150,866,217–150,866,276, GRCm38/mm10) and was strongly enriched across all ages and observed to increase with expression with age (Supplementary Data 1 and Supplementary Table 3). We validated its strong enrichment (but not increasing expression with age) by qPCR (Fig. 8b). Next, we constructed novel miRNA–mRNA regulatory networks, filtering for strongly negative correlations between miRNA–mRNA pairs ($R^2$ > 0.8, FDR < 0.05). In the absence of experimental evidence for novel miRNA binding interactions, we calculated predicted interactions using the hybridisation tool RNAhybrid, filtering for those that were predicted to bind with minimum free fold energy <25 kcal/mol ($P$ < 0.05)[33]. Fourteen miRNA–mRNA interactions passed these criteria, all of which included the top novel miRNA candidate chr3_29427 (Fig. 8c).

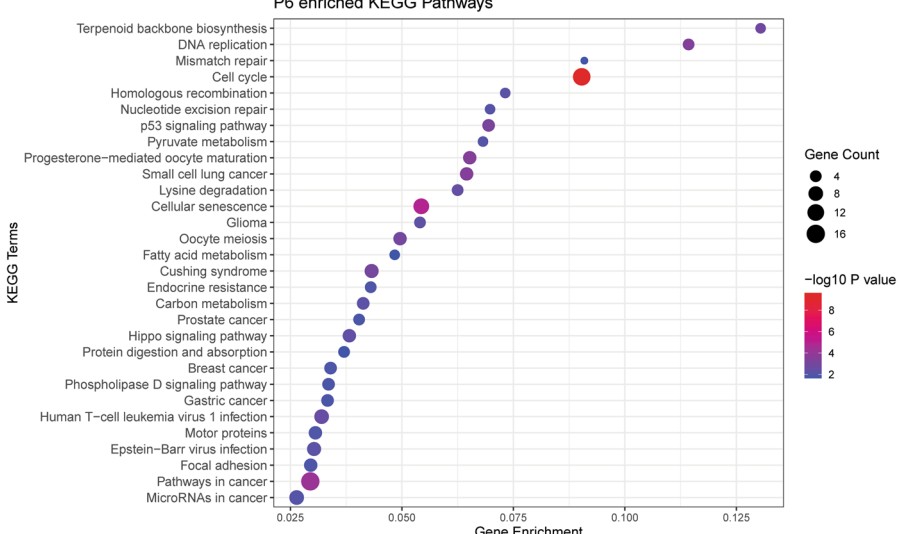

**Fig. 4 miRNA–mRNA coregulatory network highlights important CNS processes regulated by P6 microglia. a** miRNA–mRNA regulatory networks were constructed from negative correlations ($R^2 > 0.8$, FDR < 0.05) between miRNA and mRNA expression data generated from P6 microglia and validated with external evidence for interaction. miRNA(s) highlighted brown are unique to the P6 network. **b** KEGG enrichment dot plots were generated from target genes in the network. Top 30 pathways/terms (by *P* value) are represented in each plot and ordered by Gene Enrichment score (number of genes in the dataset represented in the gene set divided by the total number of genes in the gene set).

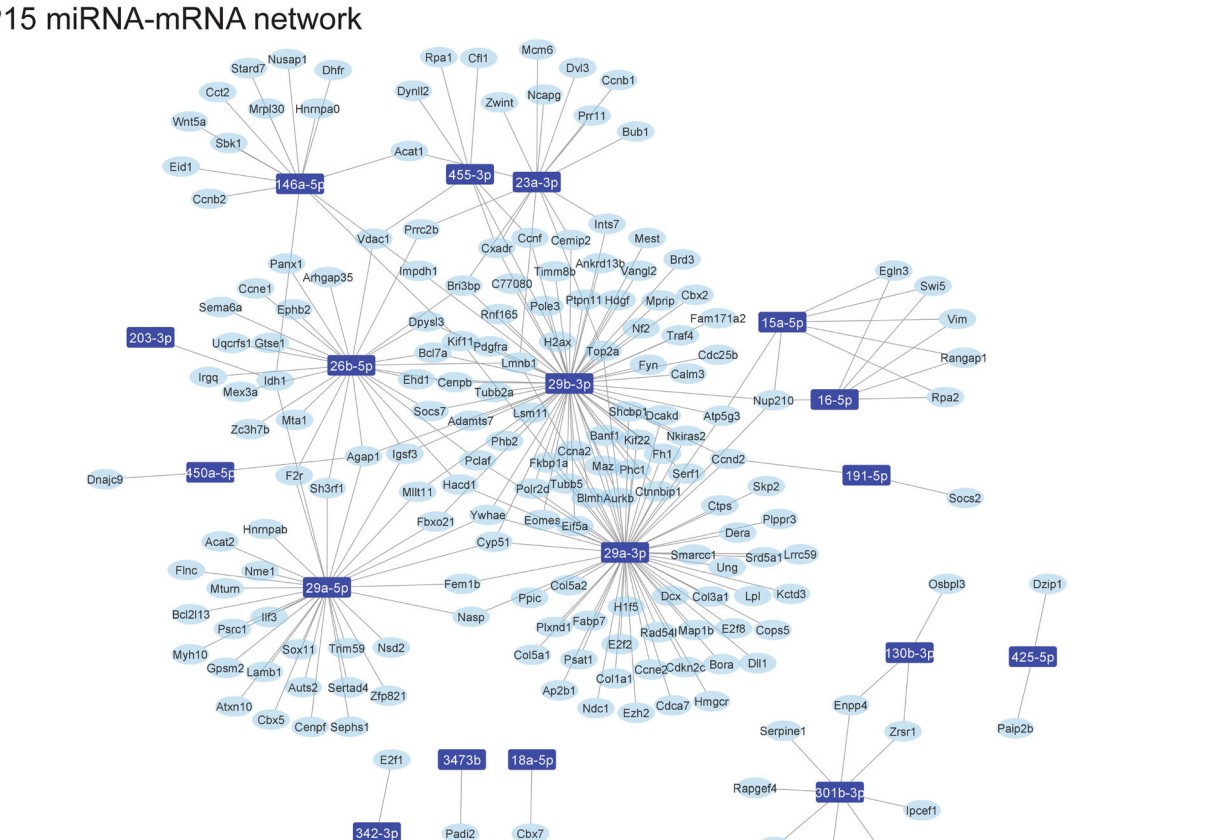

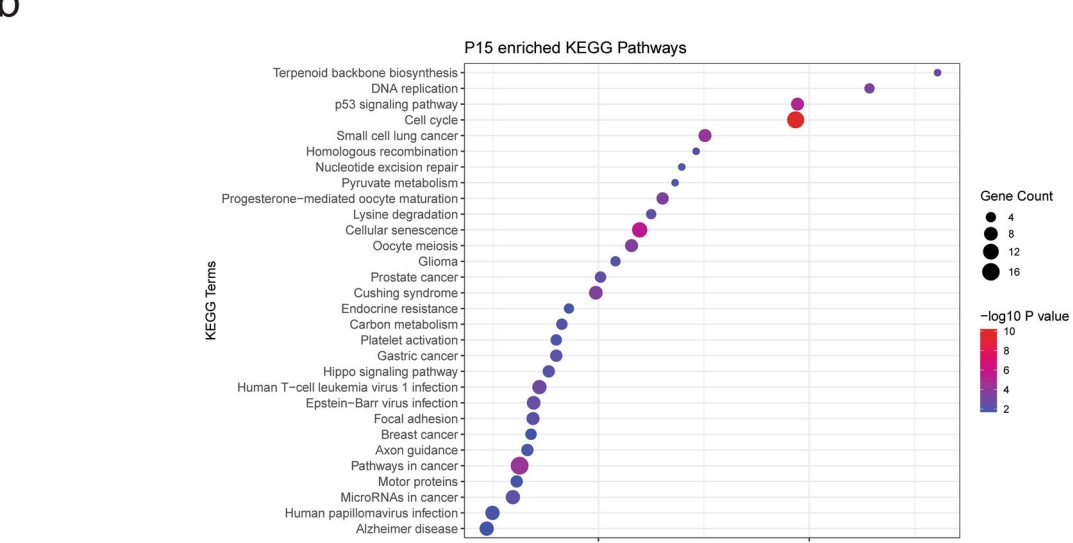

**Fig. 5 miRNA–mRNA coregulatory network highlights important CNS processes regulated by P15 microglia. a** miRNA–mRNA regulatory networks were constructed from negative correlations ($R^2 > 0.8$, FDR < 0.05) between miRNA and mRNA expression data generated from P15 microglia and validated with external evidence for interaction. **b** KEGG enrichment dot plots were generated from target genes in the network. Top 30 pathways/terms (by $P$ value) are represented in each plot and ordered by Gene enrichment score (number of genes in the dataset represented in the gene set divided by the total number of genes in the gene set).

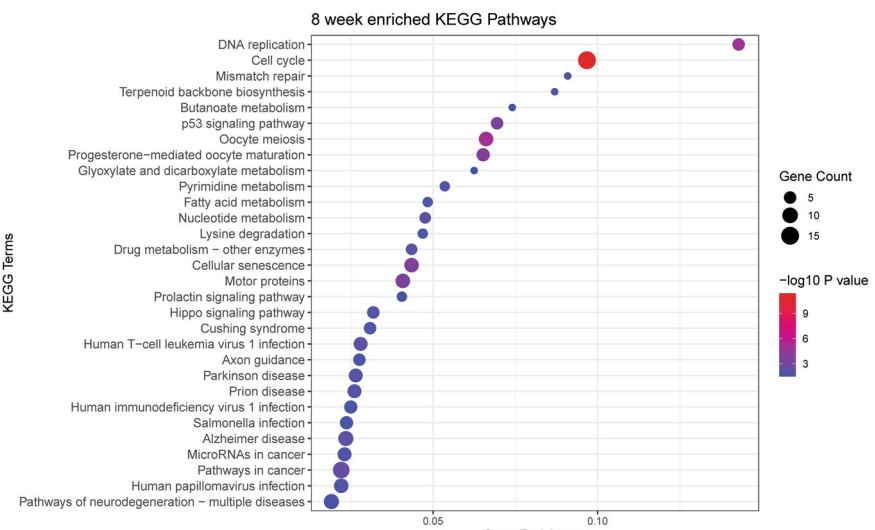

**Fig. 6 miRNA–mRNA coregulatory network highlight important CNS processes regulated by 8-week microglia. a** miRNA–mRNA regulatory networks were constructed from negative correlations ($R^2 > 0.8$, FDR < 0.05) between miRNA and mRNA expression data generated from 8-week microglia and validated with external evidence for interaction. miRNA(s) highlighted brown are unique to the 8-week network. **b** KEGG enrichment dot plots were generated from target genes in the network. Top 30 pathways/terms (by *P* value) are represented in each plot and ordered by Gene enrichment score (number of genes in the dataset represented in the gene set divided by the total number of genes in the gene set).

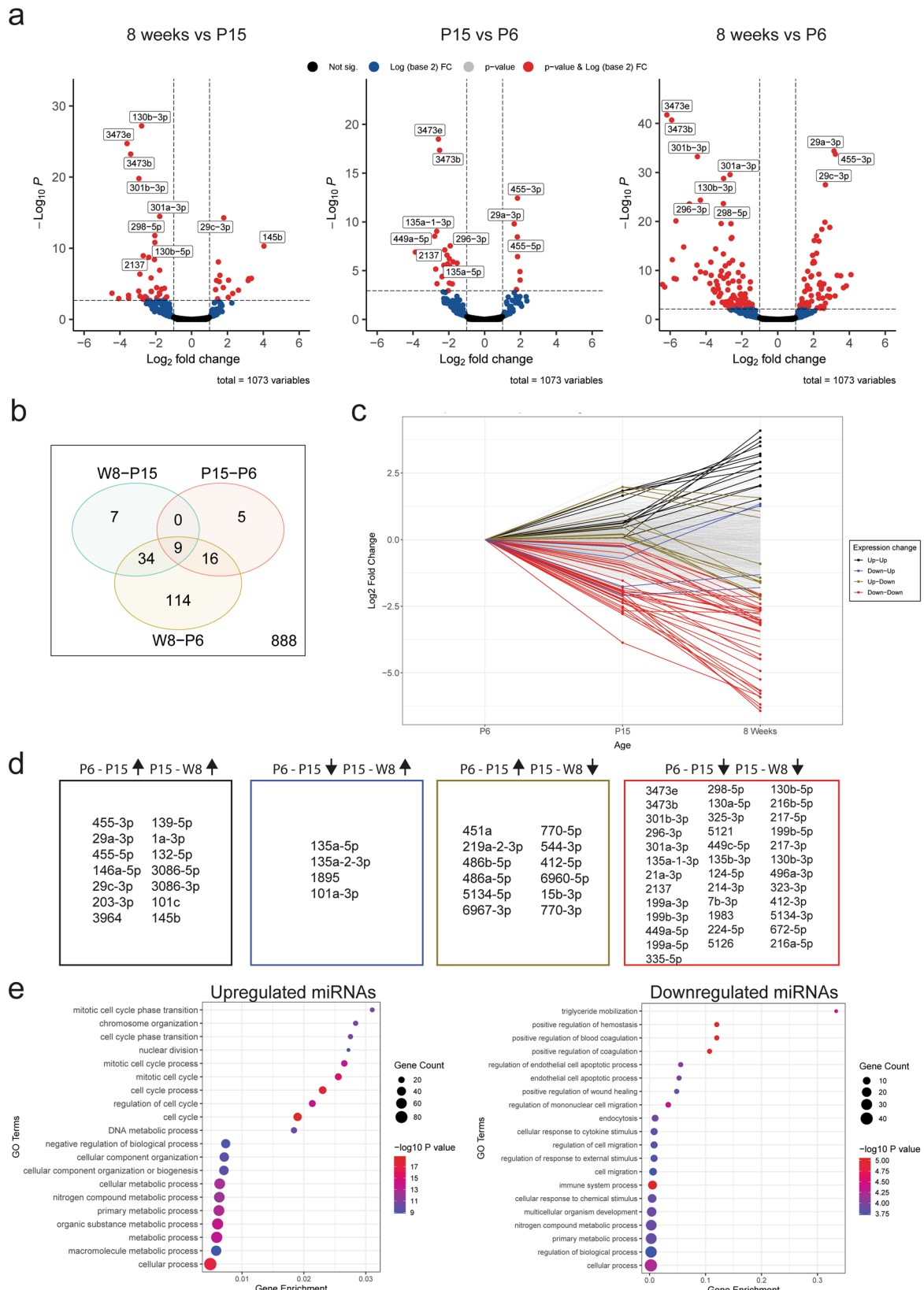

Linear modelling plotting chr3_29427 expression against the expression of two predicted targets in this regulatory network; *Birc5* and *Cdca5* highlights a consistent pattern of increasing chr3_29427 expression associated with downregulation of target genes to a negligible level of expression in 8-week microglia (Fig. 8d).

**Male and female developmental microglia exhibit no differential miRNA expression and a small difference in mRNA expression.** Mouse cohorts in this study were balanced for sex based on previous reports of sexual dimorphism in microglial genetics[34–36]. However, comparisons of miRNA and mRNA expression between sexes within each age group revealed no

**Fig. 7 Subsets of microglial miRNAs are differentially expressed over developmental time. a** Volcano plots of differentially expressed miRNAs of three pairwise comparisons for P6, P15 and 8-week (adult) microglia populations. Top ten miRNAs (by *P* value) are labelled in each plot. **b** Venn diagrams comparing differentially expressed (absolute logFC >1, FDR < 0.05) miRNAs from pairwise comparisons of age groups. One hundred and eighty-five miRNAs were differentially expressed in at least one comparison. **c** miRNA expression ($n = 1073$) was plotted as LogFC relative to expression at P6 to observe changes over developmental time. miRNAs with at least one significant fold change between age group transitions (P6-P15 and/or P15-W8) were categorised and coloured by their pattern of expression (i.e. up–up, up–down, down–up or down–down. Unchanging genes are coloured light grey). Presence of a point at a specific age group indicates that the specific transition as statistically significant (FDR < 0.05). **d** Lists of miRNAs corresponding to each category as described in (**c**). The largest subsets were the up–up and down–down categories, with 14 and 37 miRNAs respectively. **e** Gene Ontology (GO) analysis of predicted target genes (refer to methods) of consistently upregulated (up–up) and downregulated (down–down) miRNAs (related to (**d**)). Top 20 pathways/terms (by *P* value) are represented in each plot and ordered by Gene enrichment score (number of genes in the dataset represented in the specific gene set divided by the total number of genes in the specific gene set).

differentially expressed miRNAs and only six differentially expressed mRNAs (Fig. 9a, b). To further validate the absence of sexually dimorphic miRNAs, we used qPCR to directly assess the expression of candidate miRNAs previously identified as sexually dimorphic in adult microglia[35]. None of the selected candidates exhibited differential expression between sexes in 8-week-old mice. (Fig. 9c). Conversely, six mRNAs which were differentially expressed between sexes were observed across all age groups, all of which were X- or Y-linked (Supplementary Data 6). Overall, these data suggest minimal differences between male and female microglial miRNA and mRNA expression during postnatal development[20,37].

## Discussion

In this study, we have characterised the miRNAome of developmental microglia, identifying both a core miRNA signature enriched in microglia, as well as unique age-specific subsets of miRNAs. We have constructed a miRNA–mRNA network for microglia, which implicates both previously annotated and novel miRNAs in the regulation of key neurodevelopmental processes, including neurogenesis and myelination. Unexpectedly, we did not observe sex-specific miRNA expression, along with only a handful of sex-specific mRNA transcripts, indicating that during development microglia are not necessarily inherently different between sexes. Our developmentally focused dataset provides insight into the epigenetic mechanisms regulating the transition of microglia from a proliferative and highly reactive cell population to a mature homeostatic population within the CNS.

We identified a strong miRNA enrichment signature in microglia when compared to the surrounding CNS cells, with nearly 300 miRNAs upregulated in microglia in at least one age group. This was not unexpected, as studies of microglial transcriptomes have highlighted these cells as having a unique transcriptional signature[38–40]. Unlike the other major CNS cell types (astrocytes, neurons, oligodendrocytes), microglia do not derive from neuroepithelial lineages, but instead arise from the embryonic yolk sac before migrating to and establishing a self-sufficient population in the developing CNS[41–43]. Our microglial mRNA data also reflected this distinct signature. Amongst the top expressed mRNA transcripts in this sequencing data were well-defined microglial identity genes including *Cx3cr1, Csf1r, Sparc, Tmem119* and *Hexb*[44]. Together, these results highlight a unique genetic identity of microglia compared to the surrounding CNS environment.

The present study has defined a dynamic microglial miRNA expression profile, comprised of a core signature of consistently enriched miRNAs and subsets of temporally specific miRNAs. In conjunction with integrated miRNA–mRNA network analysis, we were able to characterise age-specific domains of miRNA expression and explore their influence upon microglial mRNA transcription throughout development. Firstly, 84 miRNAs were observed to be significantly enriched in microglia at all ages

compared to other CNS cell types. Many of the highly expressed miRNAs in this subset have been previously identified in microglia, including miR-146, miR-223 and miR-142[23,45,46]. These miRNAs have defined functions in the homeostatic regulation of microglial phenotype and their dysregulation is linked to inflammation and neurodegeneration[34,47–50]. In addition to this common microglial signature, our study identified 129 miRNAs that were specifically enriched in a single age group. Although most of these miRNAs were expressed in adult microglia there was enrichment of specific miRNAs at all ages, suggesting that miRNAs could be important for regulating age-specific microglial functions.

To fully characterise the roles of miRNAs in cellular systems it is important to define the gene networks which they regulate. miRNA–mRNA networks in this study were constructed using both stringent correlation thresholds and experimental validation of miRNA–mRNA interactions. Pathway analysis of target mRNAs in age-specific networks were strongly enriched for cell cycle-related processes. Developmental microglia proliferate as they colonise the brain. This includes a wave of proliferation that peaks during the second postnatal week of development, before numbers stabilise in adulthood[31]. Network analysis identified numerous miRNA candidates that could provide important regulatory influence on the microglial cell cycle. For example, the miR-301/130b cluster was identified as strongly enriched in P6 and P15 microglia. This cluster is associated with hematopoietic cell proliferation and migration. Targets of this cluster in microglia in our data included *Serpine1, Socs5* and *Ifi27* which are implicated in cell proliferation, motility, interferon signalling and endocytosis[51–55]. Also enriched in microglia was the miR-29 cluster (miR-29a, miR-29b), which is also associated with cell proliferation dynamics and downregulation of cyclins and cyclin-dependent kinases[52,56]. Another interesting candidate for the regulation of microglial development is miR-10a-5p. Previously uncharacterised in microglia, miR-10a-5p, is uniquely enriched in adult mice and is predicted to target and therefore potentially inhibit *Mcm5*, an established regulator of microglial cell cycle that is strongly expressed during embryonic and early postnatal microglial development[21,38].

In addition to cell cycle regulation, pathway analysis of miRNA–mRNA networks revealed an overrepresentation of processes related to CNS development, including neurogenesis, axon guidance and myelination. Microglia are essential modulators of these processes and their dysregulation is a major contributor towards neurodevelopmental disorders[57]. Amongst neurogenesis related genes represented in miRNA–mRNA networks was *Dypls3* (targeted by miR-26b-5p, miR-152-3p and miR-29b-3p) a cytoskeleton associated gene that is strongly enriched in amoeboid microglia during the first week of postnatal development[7,58,59]. Knockdown of *Dypls3* inhibits microglial phagocytosis and migration and therefore could be important for maintaining a specific microglial phenotype associated with

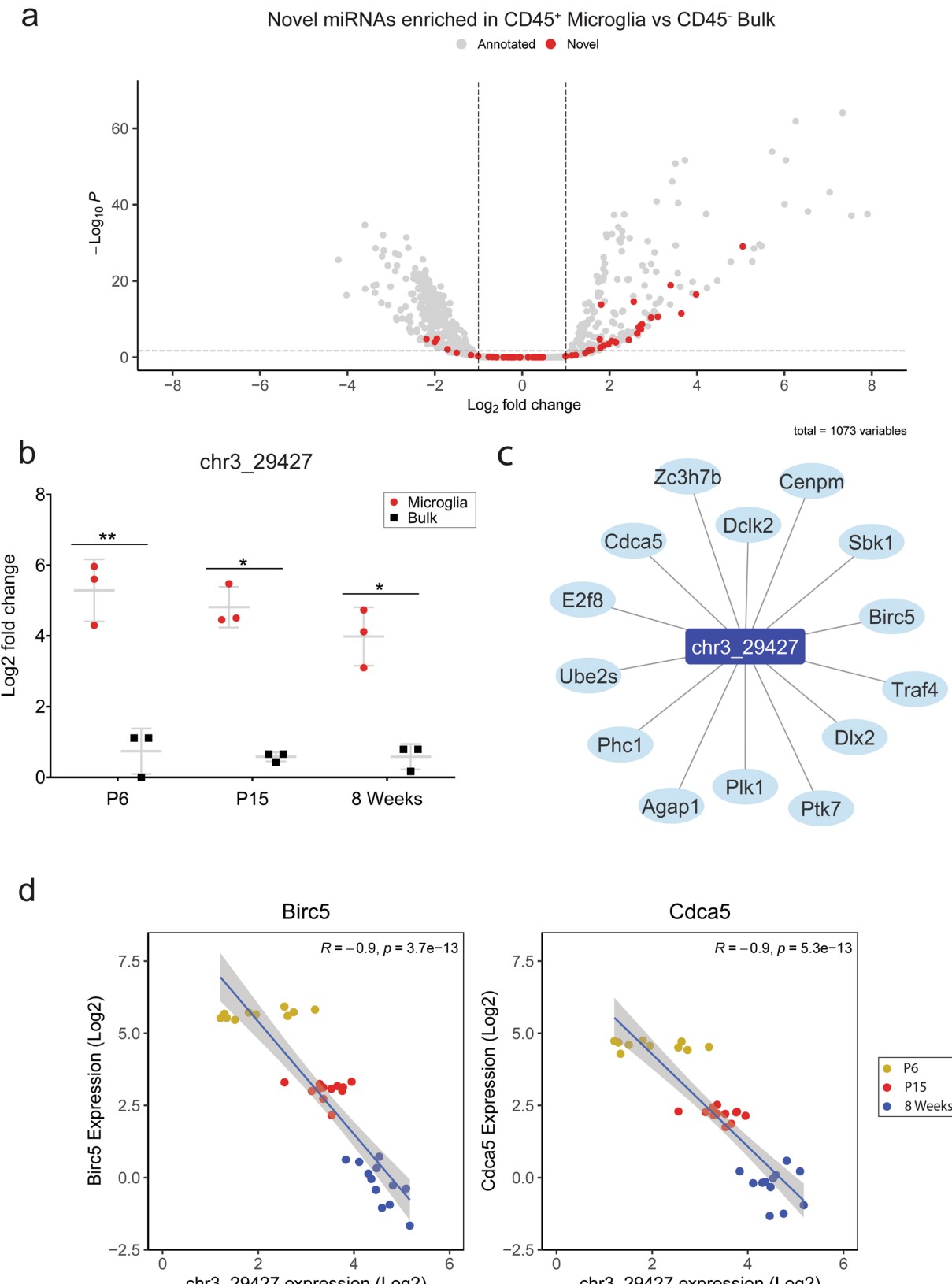

postnatal neurogenesis[58]. Microglia also regulate developmental myelination and maintain white matter integrity[2,3,60]. We identified several interesting miRNA–mRNA interactions that may mediate these processes. This included miR-29a-3p targeting *Hmgcr*, a regulator of cholesterol homeostasis that is enriched in white matter-associated microglia during myelinogenesis[61].

Another interesting candidate is miR-3473b, represented in both the P6 and P15 networks that targets a single gene, *Padi2*. PADI2 regulates citrullination of myelin basic protein (MBP), which alters myelin ultrastructure. Of note, it has been hypothesised that myelin modifications of this nature could be a source of autoimmune pathogens potentially implicated in the

**Fig. 8 miRNA discovery pipeline identified 24 novel miRNA candidates enriched in developmental microglia. a** Volcano plot of differential miRNA expression between pooled microglia and CD45$^{-ve}$ bulk samples, labelled as either novel (red) or annotated (grey) (related to Fig. 2a). Of the 54 novel miRNAs detected in all samples, 24 were identified as generally enriched in microglia. **b** qPCR expression analysis of chr3_29427 in microglia and CD45$^{-ve}$ bulk samples across developmental time ($n = 3$ males in each group). Significant enrichment was observed at all ages. Data were analysed by paired $t$ tests comparing chr3_29427 expression between microglia and CD45$^{-ve}$ bulk in each age group. *$P < 0.05$, **$P < 0.01$. **c** chr3_29427-mRNA regulatory network constructed based on strong negative correlations ($R^2 > 0.8$, FDR $< 0.05$) from mRNA expression data and prediction of interaction using RNAHybrid. Fourteen putative targets were identified for chr3_29427. **d** Log2 expression of chr3_295427 and two of its putative targets: Birc5 and Cdca5. qPCR data are presented as mean ± SD.

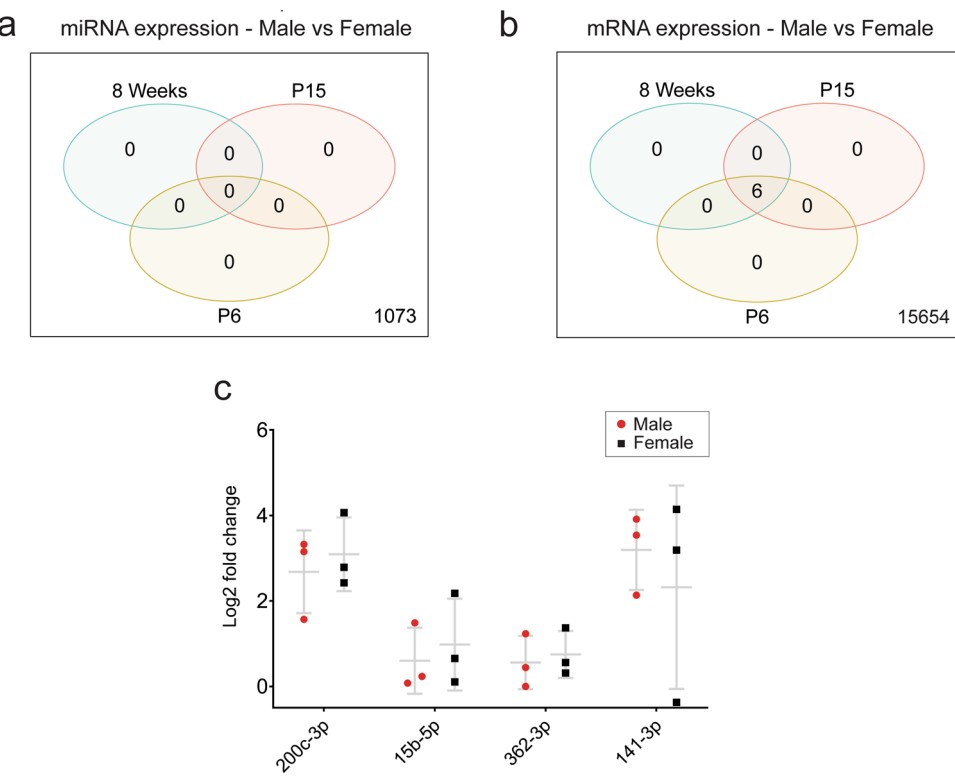

**Fig. 9 Differential miRNA and mRNA expression between male and female microglia across developmental age. a** Venn diagram comparing differential expression of miRNAs between male and female microglia in each age group. No miRNAs were differentially expressed at any age (FDR $< 0.05$). **b** Venn diagram comparing differential expression of mRNA between male and female microglia in each age group. Six mRNAs were identified as differentially expressed in every age group. **c** qPCR expression of selected miRNA candidates between male and female microglia ($n = 3$ in each group). Data were analysed by unpaired $t$ tests comparing microglia and CD45$^{-ve}$ bulk group expression for each tested miRNA. No statistically significant differences in expression were observed for any of the tested miRNAs. qPCR data is presented as mean ± SD.

pathogenesis of MS[62]. *Padi2* is enriched in subsets of microglia isolated from human MS tissue and therefore miR-3473b could play an important role in refining *Padi2* expression, both during development and in demyelinating contexts[19].

More broadly, pathway analysis of targets in all miRNA–mRNA networks identified a significant enrichment of pathways related to inflammatory signalling pathways and immune activation. This included PI3k-Akt and JAK-STAT signalling, central regulators of microglial mediated neuroinflammation[63,64]. We also observed enrichment of miRNAs with links to inflammation and neurodegeneration which are not well defined in microglia including miR-511-3p, miR-1895 and miR-6983-5p. miR-511-3p has been characterised in tumour and lung associated macrophage populations, encoded by and co-regulated with the *CD206* gene, where it is associated with anti-inflammatory (M2-like) activation. miR-6983 is downregulated in oligodendrocytes during experimental autoimmune encephalomyelitis (EAE), an animal model of demyelination, and is associated with disrupted ferroptosis pathways that are implicated in demyelination and neurodegeneration[65]. With functions related to macrophage polarisation and pathways of

neurodegeneration, further work is required to elucidate specific roles for these highly expressed miRNAs in microglia. Furthermore, we acknowledge that the networks we have identified are based on in silico correlations, which may not necessarily represent genuine interactions in cells and tissues. In vitro and in vivo experiments will be crucial to support our identified networks.

In addition to previously annotated miRNAs, we identified 24 novel miRNA candidates enriched in microglia. Many of these, including the highly expressed chr3_29427, may regulate important microglial functions in the CNS. Amongst putative mRNA targets of chr3_29427 were key regulators of cell cycle and neurogenesis, including *Cdca5* and *Birc5*. *BIRC5* has been previously characterised as highly expressed by proliferating microglia during development and in response to injury[66]. These predicted novel candidates may be critical regulators of unique microglial biology and warrant future functional validation in vitro and in vivo.

We did not identify any association of sex with microglial miRNA abundance. Specifically, 6 mRNAs but no miRNAs were identified as differentially expressed between male and female

microglia across development. There is growing evidence to suggest that sex hormones influence the developmental trajectories of male and female microglia manifesting in differential regional densities, morphology, phagocytic capacity and metabolic influx[67]. Importantly, these innate differences are exacerbated during immune challenge and stress and are suggested to be a prominent influence in observed sex biases of neurological disease[34,36,68,69]. Transcriptomics studies have defined subsets of mRNAs and miRNAs that are differentially expressed between male and female microglia, in steady state and particularly following immune challenge in adult mice[34,35,68–70]. In contrast, we did not observe any differentially expressed miRNAs, a finding which was further corroborated with qPCR analysis of four candidates that were previously identified as sexually dimorphic at baseline in 6-month-old B6C3F1 mice[35]. Conversely, our study identified 6 differentially expressed mRNA transcripts across development, all of which are located on sex chromosomes and specifically differentiated in adult mice. The robust and predictable differentiation of sex-linked genes including Xist and Ddx3y support the veracity of the dataset.

What might be causing this discrepancy? Interestingly, while functional and structural differences between male and female microglia have been consistently observed, transcriptomic results are less congruent[67]. Firstly, age seems to be an important factor. A comprehensive developmental study by Hanamsagar et al. (2017) observed that transcriptionally, male and female microglia are quite similar until early adulthood (P60). Other reported transcriptional differences studies come mainly from adult mice, (12 weeks and older) reinforcing the idea that the effects of sex are more prevalent with age[34,68,70]. However, our dataset and others have reported a minimal effect of sex even in young adult cells (8 weeks)[16,20]. Further studies that assess transcription at expanded timepoints (beyond 8 weeks of development) will assist in an accurate representation of the effect of sexual development on microglial gene expression. Another hypothesis for these discrepancies is the method of microglial isolation. Microglial cells are dynamic and respond quickly to changes in their local environment[15]. While an unavoidable caveat, it has been shown that different methods of isolation (ie. immunopanning, FACS, magnetic bead sorting) can differentially alter the activation status of microglia and hence gene expression, potentially contributing to differing reports of baseline sexual dimorphism in studies of microglia[40]. In addition, aspects of sexual dimorphism in microglial gene expression are region-dependent[36,68]. Sexually dimorphic region-specific signatures would be diluted in the current study, given that the microglial populations were derived from the whole brain. Nevertheless, this result adds another perspective to the growing discussion of sexual dimorphism of microglia and suggests that sex may have a lesser influence on microglial transcription than previously reported. Specifically, dimorphism may be more prevalent post-activation, whether induced by isolation, immune challenge or in ageing-related neurodegeneration.

In summary, we characterised a unique and highly controlled dataset of miRNA expression in developing microglia, capturing a dynamic profile that may contribute to the complex roles that microglia adopt in postnatal development and the establishment of homeostasis. This miRNAome is composed of both a relatively unchanging subset of highly enriched miRNAs and subsets of miRNAs that exhibit age-dependent expression profiles. Correlation of known and novel miRNAs with mRNA expression data enabled the identification of target mRNA networks that mediate neurodevelopmental and immune processes that are central to microglial function in the CNS. In combination with environmental cues, epigenetic mechanisms shape the genetic landscape of microglia and refine phenotypes. Dysregulation of these epigenetic mechanisms can be a major driver of neurodevelopmental and neurodegenerative disorders. This developmental profile of paired miRNA and mRNA expression represents an important step in understanding the genetic regulation of microglia and their impact on the CNS in health and disease.

## Methods

**Experimental and analytical design.** The experimental paradigm was designed to identify miRNAs expressed specifically in microglia compared with other CNS cells across three developmental timepoints; P6, P15 and 8 weeks. In addition, all groups contained an equal number of male and female mice to allow for the identification of any sex-specific miRNAs. An overview of the experimental and analytical design is given in Fig. 1.

**Chemicals and reagents.** All chemicals were purchased from Sigma (CA, USA) and all cell culture media were purchased from Gibco (TX, USA) unless otherwise stated.

**Isolation of microglial populations from murine brain tissue.** All animal experiments were approved by the animal ethics committee at the Florey Institute of Neuroscience and Mental Health and conducted in accordance with the National Health and Medical Research Council Guidelines. The whole brain was extracted from male and female C57Bl/6 mice at appropriate ages (Fig. 1). Microglia and CD45$^{-ve}$ bulk cell populations were isolated using immunopanning (adapted from refs. [71,72]).

Following brain collection, meninges and blood clots were manually removed with forceps and brain tissue was processed into small (< 1mm$^3$) pieces. Tissue was incubated in 10 ml papain buffer (20U/ml Papain, Worthington, OH, USA), 0.46% glucose, 26 mM NaHCO$_3$ 0.5 mM EDTA, 1× EBSS) and 200 µl DNase I (0.4%, 12,500 U/ml, Worthington) at 34 °C for 100 min with a consistent CO$_2$ flow and periodic shaking.

Digested tissue was firstly washed and then gently triturated in low ovomucoid protease inhibitor solution (0.015 g/ml BSA (Sigma), 0.015 g/ml trypsin inhibitor (Worthington) in DPBS at pH of 7.4) to form a single-cell suspension. High ovomucoid solution (0.030 g/ml BSA (Sigma), 0.030 g/ml trypsin inhibitor (Worthington), in DPBS at pH of 7.4) was gently pipetted to form a layer underneath the cell suspension and cells were spun down for 15 min at 250×g. The cell pellet was resuspended in 0.02% BSA solution and filtered through a nitrex mesh to remove cell clumps and debris. Pre-prepared immunopanning dishes were set up as follows: a 10-cm Petri dish was incubated at 4 °C overnight with 30ul anti-rat antibody (Jackson ImmunoResearch Laboratories, Inc, 112-005-167) in 10 ml 50 mM Tris-HCL (pH 9.5). Following overnight incubation the plate was washed three times with DPBS and then incubated with 20 µl anti-CD45 antibody (BD Pharmingen, catalogue #550539) in 12 ml of 0.2% BSA for a minimum of 2 hours. Plates were washed three times with DPBS and single-cell suspensions were added to pre-coated plates and incubated for 30 min with periodic shaking. Unbound cells were collected, spun down and then resuspended in 700 µl Qiazol. This population of microglial-depleted CNS cells was referred to as 'CD45$^{-ve}$ bulk' and used as a comparison population for downstream analysis. Bound cells (CD45$^{+ve}$ microglia) were scraped off the plate into 700 µl Qiazol for further processing.

**Library construction and RNA sequencing.** Total RNA was extracted using the miRNeasy mini kit (Qiagen) in accordance with manufacturer's protocol. The quality of RNA samples was assessed using a bioanalyzer. Samples with an RNA integrity score (RIN) greater than 7 were used for library construction. One sample did not pass RNA quality control and so was excluded from library preparation. mRNA libraries were prepared using MGIEasy RNA Directional RNA library prep chemistry (MGI), and small RNA libraries were prepared using the MGIEasy Small RNA library prep chemistry (MGI). Libraries were pooled and sequenced across four lanes of MGISEQ2000-RS sequencing technology, yielding approximately 400 M reads per lane (2 lanes for miRNAseq (50 bp single end), 4 lanes for mRNAseq (paired-end 100 bp)). Library construction and sequencing were performed by personnel at Micromon Genomics (Monash University, Melbourne, Australia).

**Preprocessing and genome alignment of miRNA/mRNA sequencing data.** Adapter sequences were removed from small RNA sequencing data, and the remaining reads were filtered by size (> 18 bp) using fastp (ver 0.20.0). Trimmed reads were mapped to mature miRNAs from miRbase (https://www.mirbase.org), and the remaining reads were mapped to the mouse reference genome (mm10/GRCm38) using mirDeep2 with default settings (ver 0.1.3)[32]. Candidate novel miRNAs detected by miRDeep2 with a miRDeep2 score >4 and randfold P value < 0.05 were included in downstream analysis. mRNA sequencing data were mapped to the mouse reference genome (mm10/GRCm38) using STAR with default settings (ver 2.7.3). Gene read counts were generated using the featureCounts pipeline in Subread with the following flags (-T 8 -s 2 -p -a) (ver 2.0.0).

**Table 1 Primer sequences for mRNA qPCR.**

| Gene | Forward | Reverse | Amplicon size (bp) |
|---|---|---|---|
| TMEM119 | GTGTCTAACAGGCCCCAGAA | AGCCACGTGGTATCAAGGAG | 110 |
| CX3CR1 | CAGCATCGACCGGTACCTT | GCTGCACTGTCCGGTTGTT | 65 |
| CD45 | GAACATGCTGCCAATGGTTCT | TGTCCCACATGACTCCTTTCC | 71 |
| GFAP | TCGGTCCTAGTCGACAACTG | CAGGGAGTGGAGGAGTCATT | 69 |
| AQP4 | ACATGGAGGTGGAGGACAAC | CTTCTCCACGGTCAATGTCA | 96 |
| OLIG2 | GATGCGAAGCTCTTTGTTCA | GGCAACGACACAGAAAGAAA | 71 |
| SOX10 | TCACGACCCCAGTTTGACTA | AAGGCTGAATAGAGGCCAGA | 89 |
| CLDN5 | CTCTCAGAGTCCGTTGACCA | CCAACTGCGCATAGAAGAAA | 99 |
| ESAM | GGACAAAAGGGAGGAAATGGA | GAAGCAGGGAGGGAAGTGAAG | 69 |
| SYT1 | AGCATGTTCTGAGCAGCATC | CTGGAAGAGTGAAGCCATGA | 113 |
| NEFL | CTGCCAAGGATGAGTCTGAA | CACCCTCACCACCTTCTTCT | 54 |
| MCM5 | GCTGGGATCACTACCACCTT | TCCTCCCCTTTTGTCTCATC | 92 |
| CCNF | CTTTCTGTTGGGGACATCCT | CCCTTTCAAAGAGCTTCAGG | 142 |
| CFL1 | AGACTGTGGACGACCCCTAC | TCCTTCTTGCTCTCCTTGGT | 106 |
| SERPINE1 | GGATCGAGGTAAACGAGAGC | GAGATGACAAAGGCTGTGGA | 58 |
| PADI2 | ATCCTGCTGGTGAATTGTGA | CGCAGGATCATCTGAGACAT | 113 |

**Count normalisation and differential expression analysis.** Normalisation and differential expression analysis of miRNA and mRNA was performed using limma (ver 3.48.1, 10.18129/B9.bioc.limma) and edgeR (ver 3.34.0, 10.18129/B9.bioc.edgeR). Based on multidimensional scaling (MDS) analysis one mouse was identified as an outlier, and its miRNA/mRNA data was removed from downstream analysis. miRNAs were filtered for a minimum expression of 1 count per million (CPM) (for miRNA), and 0.5 CPM (for mRNA) in at least five samples. Gene counts were normalised using the TMM normalisation method (edgeR) and linear modelling was performed with limma-voom. All regression analyses were set up as mixed effects models including age, sex, and cell type (microglia or CD45$^{-ve}$ bulk) as fixed effects and mouse ID as a random effect[73]. The general enrichment analysis was performed by direct comparison of miRNA expression between all microglia and all CD45$^{-ve}$ bulk samples. Specific age enrichment analyses compared only the microglia and CD45$^{-ve}$ bulk samples within a relevant age group. Pairwise analyses of microglial miRNA expression were performed by direct comparison of miRNA expression between microglial age groups. Sex-specific analyses were performed by comparing male and female microglia samples within each age group. For all differential expression analyses, differentially expressed genes were identified using the TREAT method ($t$ tests relative to a threshold), set at a minimum threshold of a $\pm 2$-fold (minimum absolute $\log_2$FC of 1) change in expression with a false discovery rate (FDR) < 0.05[74]. Volcano plots were generated using EnhancedVolcano (ver 1.10.0, 10.18129/B9.bioc.EnhancedVolcano). Supplementary Tables 1–3 were generated with gt (ver 0.3.1) (https://cran.r-project.org/web/packages/gt/index.html).

**miRNA target analysis and miRNA–mRNA regulatory network generation.** To infer miRNA–mRNA interactions from gene expression data, Pearson correlations were calculated from normalised counts for all pairs of miRNA–mRNA using psych (ver 2.16, https://CRAN.R-project.org/package=psych). Only those correlations that were strongly negative and significant ($R^2 > 0.8$, FDR < 0.05) were retained for downstream analysis.

To further validate significant negative correlations, miRNA–mRNA interactions were screened for experimental evidence of interaction in any cellular system using multimiR (ver 1.140, 10.18129/B9.bioc.multiMiR) with default settings. For novel candidate miRNAs, mRNA target binding prediction was performed using RNAhybrid (ver 2.2.1, https://bibiserv.cebitec.uni-bielefeld.de/rnahybrid), whereby novel miRNA sequences were aligned with 3'-UTR mouse sequences (ENSEMBL, release 104) using the following parameters (-e -25 -p 0.05).

miRNA–mRNA interactions which were identified as negatively correlated from expression datasets and validated by multimiR or RNAHybrid were included in downstream pathway analyses and network construction. Visualisations of miRNA–mRNA networks were generated using Cytoscape (https://cytoscape.org/). KEGG (Kyoto Encyclopedia of Genes and Genomes) and GO (gene ontology) enrichment analysis were performed on the target mRNAs in generated networks using the kegga and goana modules in limma, respectively (ver 3.48.1, 10.18129/B9.bioc.limma).

**Assessment of cellular proportions.** Cellular proportions were inferred from bulk immunopanned RNA-seq gene expression matrices using CIBERSORTx High-Resolution Docker container[75]. To create a reference signature matrix, we used a single-cell RNA-seq dataset (scRNA-seq) of CD45 and Cx3cr1 FACS sorted cells throughout the mouse lifespan (E14.5, P4/P5, P30, P100 and P540)[20]. Crucially the authors identified both microglia subpopulations and a small population of intermixed monocytes and macrophages. CIBERSORTx currently recommends user input of no more than 5000 different single-cell profiles when creating the

signature matrix. To meet this recommendation, we randomly downsampled the scRNA-seq dataset to 5000 cells using the 'sample' command in R with the seed set to 40[76]. After creation of the reference matrix, CIBERSORTx was run with 'S-mode' batch correction, filtration to immune relevant genes, faction set to 0.25, and sampling set to 1.

**qRT-PCR of mRNA expression.** In all, 100 ng of total RNA isolated from microglia/ CD45$^{-ve}$ bulk samples was reverse transcribed using Taqman reverse transcription reagents (Invitrogen) in accordance with the manufacturer's instructions. In total, 25 μl qPCR reactions were prepared as follows; 12.5 μl 2× SYBR green master mix (ABI Biosystems, MA, USA), 1 μM of forward and reverse primers and DNAse free water. Primer sequences are outlined in Table 1. Expression of 18 S was used to normalise for RNA input. Relative fold change of expression was calculated using the 2(-delta-delta C(T)) method and log transformed[77].

**qRT-PCR of miRNA expression.** Overall, 50 ng of total RNA isolated from microglia/ CD45$^{-ve}$ bulk samples was reverse transcribed using the miRCURY LNA RT kit (Qiagen) in accordance with the manufacturer's instructions. The miRCURY LNA miRNA PCR assay system (Qiagen) was used to detect specific miRNA expression. Expression of ribosomal 5 S was used to normalise for RNA input for all samples. Relative fold change of expression was calculated using the 2(-delta-delta C(T)) method and log transformed[77]. Paired $t$ tests were performed to identify consistent expression differences between paired microglia/ CD45$^{-ve}$ bulk samples for each miRNA. Unpaired $t$ tests were performed to identify differences in candidate miRNA expression between male and female microglia subgroups. Statistical analyses were performed using GraphPad Prism (ver 9.2.0, Graphpad, CA, USA).

**Statistics and reproducibility.** RNA-seq sample size was determined to match or exceed those of similar studies[34,35], such that comparisons between age groups were 12 vs 12 (matched for sex), and between tissue types were 36 vs 36 (correcting for animal, age and sex). For differential expression analysis, we used the empirical Bayes procedure implemented in R limma software to maximise sensitivity to detect differentially expressed genes. To prioritise differences most likely to be biologically relevant, we subsequently applied the limma treat procedure to discard results with an absolute fold change <2. Statistical analyses are described within each relevant section.

**Reporting summary.** Further information on research design is available in the Nature Portfolio Reporting Summary linked to this article.

## Data availability
The data that support the findings of this study are openly available in the gene expression omnibus (GEO) at [https://www.ncbi.nlm.nih.gov/geo/], with accession numbers GSE229977 (mRNA sequencing) and GSE229981 (miRNA sequencing). All data underlying graphs and charts have been included as Supplementary Data 7.

## Code availability
The code used to analyse and generate datasets is available upon request.

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

## Acknowledgements

The authors would like to acknowledge the support of the Australian Government Research Training Program Scholarship (A.D.W.), National Health and Medical Research Council (APP1175775; T.J.K. and APP1184421; S.F.), Multiple Sclerosis Australia (21-6-008; T.J.K. and 21-3-038; S.S.), the Brain Cancer Centre (S.F.) and Carrie's Beanies 4 Brain Cancer (S.F.). The Florey Institute of Neuroscience and Mental Health and the Walter and Eliza Hall Institute of Medical Research acknowledge the strong support from the Victorian Government, particularly the funding from the Operational Infrastructure Support Grant.

## Author contributions

Conceived and designed the experiments: M.D.B., T.J.K., A.D.W. and B.R.E.A. Performed the experiments: A.D.W., S.S., A.A. and M.D.B. Analysed and interpreted the data: A.D.W., S.S., S.F., T.J.K., B.R.E.A. and M.D.B. Wrote the paper: A.D.W., M.D.B. and B.R.E.A.

## Competing interests

The authors declare no competing interests.
