## [Peer Review File · Communications Biology]

Reviewers' comments:

Reviewer #1 (Remarks to the Author):

Microglia, the brain-resident macrophages, are key to the development and homeostasis of the central nervous system (CNS). In this regard, aging's contribution to microglia dynamics is of great interest. The manuscript by Walsh and collaborators explores the miRNA-mRNA network landscape during central nervous system neonatal development. Caution is needed regarding the identity of the isolated cells for analysis. Nevertheless, the study highlights the relevance of the miRNA-mRNA axis in CNS development.

The following points could be addressed to enrich the discussion.

Major points:

1. Perhaps the biggest issue to be addressed in the study is the specificity of the isolated microglia. The experimental design of the study defines microglia cells as the CD45+ cells isolated by immunopanning. However, CD45+ cells include all nucleated hematopoietic cells. Also noteworthy is the fact that immune cells tend to adhere to immunopanning plates even in the absence of an antibody. Table 1 confirms the enrichment for immune cells with the presence of myeloid markers (e.g., *Csfr1*, *Ast3*, etc).

Therefore, the identified temporal miRNA signature does not necessarily reflect a specific microglia signature but an overall central nervous system CD45+ myeloid signature.

2. The lack of microglia specificity could help explain the lack of differential expression in previously identified sexually dimorphic miRNAs (Fig. 9C), which would be diluted in the myeloid population.

3. The temporal analysis of the miRNA-mRNA network during development is of extreme importance. A comparison to a pre-natal and adult homeostatic timepoint would be interesting. The reasoning behind focusing on neonatal development should be addressed in the introduction.

4. Myeloid cells are highly responsive to their microenvironment and the conditions of the isolation could affect their expression profile (Mattei, *Int J Mol Sci.* 2020;21(21):7944. Published 2020 Oct 26. doi:10.3390/ijms21217944; Ocañas, *eNeuro.* 2022;9(2):ENEURO.0348-21.2022; . Thus, the digestion with papain at 34oC for 100 min might significantly alter mRNA and miRNA expression. An enrichment analysis of the gene signature of microglia isolated with incubation steps at >30oC in the mRNA profile of the CD45+ cells would help assess the extent of the influence of this incubation step in the results.

Minor points:

1. Although stated at the beginning of the results and materials and methods, as well as Fig. 1, it is important to make clear throughout the article that the bulk cells are actually CD45- cells.

2. It would be informative to highlight either on the title or abstract that the study comprises the analysis of neonatal CNS development.

Reviewer #2 (Remarks to the Author):

The study by Walsh et al. aimed to investigate the effect of small non-coding microRNAs (miRNAs) on microglial functions during the CNS development. They sequenced microglial miRNAome and mRNAome using p6, p15 and 8 weeks old mouse brains to determine the changes during brain development and homeostasis. They have identified some known and novel miRNAs from the samples they sequenced. They then analyzed miRNA-mRNA networks related to fundamental

developmental processes, revealing a unique developmental trajectory of microglial miRNA expression during the CNS development. They, therefore, concluded that identified miRNAs are important modulators of microglial phenotype during the CNS development. Although the study is interesting, it lacks essential studies to validate the identified miRNAs in regulating the CNS development. In addition, the target proteins of identified miRNAs should also be determined and evaluated to demonstrate their functions during the CNS development. Without these essential studies to validate the observed miRNA results, the study is basically only comparing microglial miRNA expression among three age groups during the CNS development.

Reviewer #3 (Remarks to the Author):

Walsh and colleagues have done RNA-seq on miRNA and mRNA on microglia at different stages of postnatal development and looked at the changes that occur at the level of both transcript classes. They identify a number of networks containing miRNA and mRNA related to development, immunity, and disease states. My expertise in reviewing this paper is looking at miRNA and mRNA in systems as well as knowledge with microglia in disease.

On the whole, the study is well designed with no clear technical issues and I think this study would add to the field, if only because of the wealth and depth of data generated. Based on this, I would recommend that this paper is accepted provided that adequate consideration and response is provided to the below questions – as there were no page or line numbers, I have endeavoured to use Figures to highlight the corresponding part of the text.

Figure 3A: The authors in identifying enrichment of miRNA in the RNA-seq from microglia compared to bulk cells have set a threshold of two-fold in addition to the FDR p-value of 0.05. Considering the large number of miRNA that were enriched and that the top 15 miRNA enriched showed a 32-fold enrichment, did the authors consider using a higher threshold for their analysis, such as 10-fold? How would this change their analysis? Additionally, did the authors have an expression threshold as going from an average of 1 copy to 8 copies is biologically irrelevant.

Figure 3C: For the qPCR of enriched candidates, why was only these four miRNA in 8 week old mice investigated considering for an additional 5 candidates, they did all three timepoints? Were these specific to the timepoint? What about other timepoints?

Table 2: It would be useful to highlight those miRNA that are commonly enriched across all three timepoints in the Top 10 and/or overall.

Why was $r < -0.894$ chosen? These seems oddly specific.

Figure 4-6: It would be useful to highlight genes in the key pathways in the network diagram to see where they sit in these networks – this would help to highlight the interconnectedness of the networks and the involvement of the multiple miRNA converging on targeting genes in certain pathways - otherwise the usefulness of these figures is minimal. This doesn't necessarily need to be the ones top of the table, but should be at least the ones mentioned in the text.

As far as I can tell, the authors have not validated the expression of the mRNA in the samples with qPCR and Western – why was this not done? This should be done to validate the RNA-seq data and ensure that the conclusions that they are making are not an artifact of the RNA-seq bioinformatics. This applies to these networks and to the identified targets of the novel miRNA.

Figure 8A: Did the authors validate that the miRNA form proper stem loop sequences? In my experience, I have found novel miRNA do not form stem loops that would be processed by Dicer and Drosha and so are unlikely to be real miRNA and in fact fragments of other RNA species.

The discussion is very long (longer than the rest of the paper) and I think it should be reduced – considering this study hasn't done any work to manipulate any of these pathways or even validated that the gene targets of the miRNA at qPCR or Western level are changed, there is an overly large amount of conjecture and some of the conclusions in my mind have been

overextended. While I recognise that they have selected high thresholds for the generation of their networks, my experience has shown me that this does not guarantee results.

Reviewer 1		
Reviewer comment	Response	Summary of changes made
1. Perhaps the biggest issue to be addressed in the study is the specificity of the isolated microglia. The experimental design of the study defines microglia cells as the CD45+ cells isolated by immunopanning. However, CD45+ cells include all nucleated hematopoietic cells. Also noteworthy is the fact that immune cells tend to adhere to immunopanning plates even in the absence of an antibody. Table 1 confirms the enrichment for immune cells with the presence of myeloid markers (e.g., Csf1, Ast3, etc). Therefore, the identified temporal miRNA signature does not necessarily reflect a specific microglia signature but an overall central nervous system CD45+ myeloid signature.	The mouse brains harvested in this project were healthy, and therefore we do not expect to observe a substantial number of CD45+ve non-microglial cells, although as we noted in the manuscript we do not expect our samples to be 100% microglial cells only. Nonetheless, we appreciate the importance of understanding more precisely the purity of our samples. To address this question we have undertaken to use single cell data derived from mouse microglia across development (Hammond et al. 2019). Importantly this dataset also included a monocyte/macrophage signature, allowing us to undertake deconvolution of our dataset to estimate the proportion of each sample which were non-microglial cells. We found that all samples were >95% microglia.	(1) Dr. Saskia Freytag is an expert in the area of deconvolution and was central to this new analysis. Dr. Freytag has now been included as an author in this manuscript. (2) The method by which we deconvoluted our samples has been included in the methods in a section entitled "Assessment of cellular proportions" (line 650) (3) The manuscript describing the single cell dataset used for deconvolution, and the manuscripts describing the software employed have been added to the references. (4) The results of the deconvolution have been added as panel B in supplementary figure 1.
2. The lack of microglia specificity could help explain the lack of differential expression in previously identified sexually dimorphic miRNAs (Fig. 9C), which would be diluted in the myeloid population.	As noted above, our samples are of a consistent high purity, with all samples >95% microglia, and thus the lack of sex-specific expression of miRNAs in our microglia cannot be attributed to reduced purity.	No changes made
3. The temporal analysis of the miRNA-mRNA network during development is of extreme importance. A comparison to a pre-natal and adult homeostatic timepoint would be interesting. The reasoning behind focusing on neonatal development should be addressed in the introduction.	miRNA during microglial development as published assessment of the mRNA transcriptome has highlighted the existence of unique developmental phenotypes, but the miRNAome had not been explored in the context of microglial development. Our specific timepoints were chosen to represent early postnatal development (P6, P15) as well as a young adult timepoint (8 weeks). We would not classify these times as "neonatal", as an adult timepoint is represented in our dataset. However, we have added a further sentence highlighting our interest in postnatal and early adult microglial development.	We have added the following sentence to the introduction on line 98 "These timepoints represent critical stages in CNS development in the mouse."
4. Myeloid cells are highly responsive to their microenvironment and the conditions of the isolation could affect their expression profile (Mattei, Int J Mol Sci. 2020;21(21):7944. Published 2020 Oct 26. doi:10.3390/ijms21217944; Ocañas, eNeuro. 2022;9(2):ENEURO.0348-21.2022; . Thus, the digestion with papain at 34oC for 100 min might significantly alter mRNA and miRNA expression. An enrichment analysis of the gene signature of microglia isolated with incubation steps at >30oC in the mRNA profile of the CD45+ cells would help assess the extent of the influence of this incubation step in the results.	As noted in our discussion, we acknowledge the influence of experimental/technical procedures on the microglial transcriptome. To address the reviewer's comments regarding temperature, we correlated differential gene expression between age groups with subsets of temperature associated genes identified in Mattei et al. (2020). This correlation was not significant across any developmental stage, indicating that observed transcriptional changes are not strongly driven by temperature.	No changes made
Minor point 1. Although stated at the beginning of the results and materials and methods, as well as Fig. 1, it is important to make clear throughout the article that the bulk cells are actually CD45-cells.	We agree this is a critical point, which we have now highlighted carefully throughout the manuscript.	References to "bulk" have been replaced with "CD45-ve bulk" throughout the manuscript

Minor point 2. It would be informative to highlight either on the title or abstract that the study comprises the analysis of neonatal CNS development.	As noted above we would not characterise this study as exclusive of neonatal CNS development, given the inclusion of data from an adult timepoint. We feel that the title and abstract fairly and specifically represent the timepoints examined.	No changes made
Reviewer 2		
Reviewer comment	Response	Summary of changes made
The study by Walsh et al. aimed to investigate the effect of small non-coding microRNAs (miRNAs) on microglial functions during the CNS development. They sequenced microglial miRNAome and mRNAome using p6, p15 and 8 weeks old mouse brains to determine the changes during brain development and homeostasis. They have identified some known and novel miRNAs from the samples they sequenced. They then analyzed miRNA-mRNA networks related to fundamental developmental processes, revealing a unique developmental trajectory of microglial miRNA expression during the CNS development. They, therefore, concluded that identified miRNAs are important modulators of microglial phenotype during the CNS development. Although the study is interesting, it lacks essential studies to validate the identified miRNAs in regulating the CNS development. In addition, the target proteins of identified miRNAs should also be determined and evaluated to demonstrate their functions during the CNS development. Without these essential studies to validate the observed miRNA results, the study is basically only comparing microglial miRNA expression among three age groups during the	Although we agree that ultimately functional studies will be required to demonstrate the contribution of identified miRNAs to microglial development, these studies are beyond the scope of this manuscript. This manuscript encompasses more than a catalogue of microglial miRNAs across development (although we note our data is the first such comprehensive dataset in this respect), we have also identified likely mRNA targets across development. Further, we have identified that the miRNAome is not necessarily differential between sexes. These data and findings stand alone, separate from future functional studies.	No changes made
Reviewer 3		
Reviewer comment	Response	Summary of changes made
1. Figure 3A: The authors in identifying enrichment of miRNA in the RNA-seq from microglia compared to bulk cells have set a threshold of two-fold in addition to the FDR p-value of 0.05. Considering the large number of miRNA that were enriched and that the top 15 miRNA enriched showed a 32-fold enrichment, did the authors consider using a higher threshold for their analysis, such as 10-fold? How would this change their analysis? Additionally, did the authors have an expression threshold as going from an average of 1 copy to 8 copies is biologically irrelevant.	Our threshold of logFC >2 enables analysis of genes/miRNAs that are meaningfully differentially expressed, giving us an overview of the transcriptome and its associated changes. We acknowledge that there are other perspectives on the threshold that may be considered meaningful. We have therefore extended the information for each of our enriched datasets provided in the supplementary data, enabling interested readers to explore the data and underlying biology with thresholds of their choice. As noted in the methods section, miRNAs were filtered for a minimum expression of 1 count per million (CPM) (for miRNA), and 0.5 CPM (for mRNA) in at least 5 samples. These are generally considered acceptable thresholds for transcript expression. Additionally, by only including miRNAs/genes that surpass these minima in at least 5 samples, we can more confidently conclude that miRNAs/genes are	Extended supplementary information (now represented in Tables S1 and S2) more information for enriched miRNAs, including predicted target information and (Gene Ontology) pathway enrichment.

2. Figure 3C: For the qPCR of enriched candidates, why was only these four miRNA in 8 week old mice investigated considering for an additional 5 candidates, they did all three timepoints? Were these specific to the timepoint? What about other timepoints?	In figure 3C we chose to investigate 4 miRNAs at only a single timepoint (8 weeks) as they were representative of miRNAs which formed a "microglia specific signature". Nonetheless, we acknowledge that a subset of these also change over time in our data set and there is value in validation of these changes. We have therefore extended the analysis of these miRNAs to P6, P15 and 12 weeks. The latter timepoint is different to our original 8 week dataset as we had insufficient RNA remaining from our original analysis.	The results from this extended validation are included in a new supplementary figure (Supplementary figure 2). The sequences of the relevant primers have been added to Table 5 in the methods.
3. Table 2: It would be useful to highlight those miRNA that are commonly enriched across all three timepoints in the Top 10 and/or overall.	We agree with this reviewer's comment and have made the appropriate changes.	Modified table 2 to highlight unique/common miRNAs expressed across developmental groups
4. Why was $r < -0.894$ chosen? These seems oddly specific.	$r < 0.984$ was chosen as it represents a negative correlation with a pearson value (R^2) of > 0.8. We agree that the presentation of this number may be slightly confusing so have amended it in the	Changed $r < -0.894$ to R^2 where appropriate, indicating that the data is filtered specifically for negative correlations
5. Figure 4-6: It would be useful to highlight genes in the key pathways in the network diagram to see where they sit in these networks – this would help to highlight the interconnectedness of the networks and the involvement of the multiple miRNA converging on targeting genes in certain pathways - otherwise the usefulness of these figures is minimal. This doesn't necessarily need to be the ones top of the table, but should be at least the ones mentioned in the text.	We agree with this reviewer's comment and have made the appropriate changes.	Figures 4,5 and 6 have been modified to highlight key miRNAs and gene targets, relevant to those mentioned in the results and discussion.
6. As far as I can tell, the authors have not validated the expression of the mRNA in the samples with qPCR and Western – why was this not done? This should be done to validate the RNA-seq data and ensure that the conclusions that they are making are not an artifact of the RNA-seq bioinformatics. This applies to these networks and to the identified targets of the novel miRNA.	We agree that validating the mRNA changes (in additon to the miRNA changes) is both reasonable and important. However, in our opinion Western analysis is both a step beyond validation and even replication, as it is measuring more than just transcription. We have therefore validated the expression of 5 genes (Mcm5; Ccnf; Cfl1; Serpine1; Padi2) which we identified as negatively correlated with key miRNAs.	The results from this extended validation are included in a new supplementary figure (Supplementary figure 3).
7. Figure 8A: Did the authors validate that the miRNA form proper stem loop sequences? In my experience, I have found novel miRNA do not form stem loops that would be processed by Dicer and Drosha and so are unlikely to be real miRNA and in fact fragments of other RNA species.	The program used to detect novel miRNAs, mirDeep2 provides predicted stem loop precursor sequences. We specifically note in the manuscript that any identified novel miRNAs need to be validated experimentally.	We have included the predicted stem loop structures for each of the microglial enriched miRNAs as a supplementary dataset, providing more context to their putative structure/functionality.

8. The discussion is very long (longer than the rest of the paper) and I think it should be reduced – considering this study hasn't done any work to manipulate any of these pathways or even validated that the gene targets of the miRNA at qPCR or Western level are changed, there is an overly large amount of conjecture and some of the conclusions in my mind have been overextended. While I recognise that they have selected high thresholds for the generation of their networks, my experience has shown me that this does not guarantee results.	We agree that that the discussion was overly long and contained speculations well beyond the current dataset. We have therefore edited the discussion to be more focussed upon direct results with more limited speculation	Discussion length has been reduced and speculation limited.
---	--	--

REVIEWERS' COMMENTS:

Reviewer #1 (Remarks to the Author):

The included supplementary material and modifications on the text better define their experimental model and limitations of the study. The authors have carefully addressed all raised concerns.

Reviewer #3 (Remarks to the Author):

I thank the authors for considering my comments about their study. They have sufficiently addressed my comments through the addition of supplementary materials, changes to tables and figures, and changes to the discussion to make it more succinct. These changes have allowed a more accurate assessment to be made about the study and will allow readers to reach their own conclusions. As such, I feel that this study can be accepted for publication.